# Benchmarking the Robustness of Spatial-Temporal Models Against Corruptions

**Chenyu Yi**[1*]  **Siyuan Yang**[1,2*]  **Haoliang Li**[3†]  **Yap-Peng Tan**[1]  **Alex C. Kot**[1]

[1]School of Electrical and Electronic Engineering, Nanyang Technological University, Singapore
[2]Interdisciplinary Graduate Programme, Nanyang Technological University, Singapore
[3]Department of Electrical Engineering, City University of Hong Kong, China
{yich0003,siyuan005}@e.ntu.edu.sg    haoliang.li@cityu.edu.hk    {eyptan,eackot}@ntu.edu.sg

## Abstract

The state-of-the-art deep neural networks are vulnerable to common corruptions (e.g., input data degradations, distortions, and disturbances caused by weather changes, system error, and processing). While much progress has been made in analyzing and improving the robustness of models in image understanding, the robustness in video understanding is largely unexplored. In this paper, we establish a corruption robustness benchmark, Mini Kinetics-C and Mini SSV2-C, which considers temporal corruptions beyond spatial corruptions in images. We make the first attempt to conduct an exhaustive study on the corruption robustness of established CNN-based and Transformer-based spatial-temporal models. The study provides some guidance on robust model design and training: the generalization ability of spatial-temporal models implies robustness against temporal corruptions; model corruption robustness (especially robustness in the temporal domain) enhances with computational cost and model capacity, which may contradict with the current trend of improving the computational efficiency of models. Moreover, we find the robustness intervention for image-related tasks (e.g., training models with noise) may not work for spatial-temporal models. Our codes are available on https://github.com/Newbeeyoung/Video-Corruption-Robustness.

## 1   Introduction

Advances in deep neural networks and large-scale datasets have led to rapid progress in both image and video understanding in the past decade. However, most of the datasets only consider clean data for training and evaluation purposes, while the models deployed in the real world may encounter common corruptions on input data such as weather changes, motions of the camera, and system errors [18][34]. Many works have shown that the image understanding models which capture only spatial information are not robust against these corruptions [10][11][18]. While some efforts have been made to improve the model robustness under such corruptions for visual content based on spatial domain, the temporal domain information has been largely ignored. Compared with images, videos contain abundant temporal information, which can play an important role in video understanding tasks. The sensitivity of human vision towards corruptions is well correlated with the temporal structure in videos [40], and the deep learning models can also benefit from temporal information besides spatial information on the generalization of video understanding [19][45]. Therefore, analyzing the robustness of deep learning models from the perspective of both spatial and temporal domain is crucial for coming out with models suitable for the real world.

---

[*]equal contribution
[†]corresponding author

35th Conference on Neural Information Processing Systems (NeurIPS 2021) Track on Datasets and Benchmarks.

In this paper, we make the first attempt to evaluate the corruption robustness of spatial-temporal models on video content by proposing to answer the following research questions.

1. how robust are the video classification models when they use temporal information?

2. how robust are the models against corruptions which correlate with a set of continuous frames, termed temporal corruptions, beyond the corruptions which only depend on content in a single frame, termed spatial corruptions (i.e., packet loss in the video stream can propagate errors to the consequent frames)?

3. what is the trade-off between model generalization, efficiency, and robustness?

Particularly, we propose two datasets (Mini Kinetics-C, Mini SSV2-C) based on current large-scale video datasets to benchmark the corruption robustness of models in video classification. We choose these two datasets as the benchmark for two reasons. Firstly, Kinetics [4] and SSV2 [14] are the most popular large-scale video datasets with more than 200K videos each. Secondly, Kinetics relies on spatial semantic information for video classification, while SSV2 contains more temporal information [5][36]. It enables us to evaluate the corruption robustness of models on these two different types of data. To estimate the unseen test data distribution under the natural circumstance, we construct these two datasets by applying 12 types of corruptions which are common in video acquisition and processing on clean validation datasets. Each corruption contains 5 levels of severity. Different from image-based benchmarks [18][21][32][44], video-based datasets have another temporal dimension, so the corruptions we propose are separated into spatial domain and temporal domain. To estimate the distribution of corruptions in nature, we reduce the proportion of noise and blur, increase the corruption type caused by environmental changes and system errors.

Based on the proposed robust video classification benchmark, we conduct large-scale evaluations on the corruption robustness of the state-of-the-art CNNs and transformers. For a fair comparison, all models are trained on clean videos while evaluated on corrupted videos. We also examine the effect of training models on corrupted data on corruption robustness. Based on the evaluation, we have several findings: from the aspect of architecture, transformer-based model outperforms CNN-based models on corruption robustness; the robustness against temporal corruptions increases with generalisation ability of spatial-temporal models; the corruption robustness increases with the computation cost and capacity of models, which contradicts the current trend of improving the efficiency of models. From the aspect of data augmentation, training models with noise or corruption degrades generalization and corruption robustness in video classification, while noise or corruption data augmentation play an important role in improving the robustness of image classification models [30][33][49].

Our contributions can be summarized as follows: 1) we propose a robust video classification benchmark for evaluating the corruption robustness of models. Apart from spatial corruptions in image-based robustness benchmarks, we take temporal corruptions into considerations in our benchmarks due to the temporal structure of videos. 2) We benchmark the state-of-the-art video classification models and draw conclusions on improving the robustness of models in the future. 3) We examine the impact of training with corruption for video classification tasks. We show that it remains an open question to improve the corruption robustness of spatial-temporal models from the perspective of data augmentation.

## 2 Related Work

### 2.1 Corruption Robustness of Neural Networks

The vulnerability of deep neural networks against common corruptions has been well studied in the past few years. For example, weather like rain, fog can cause corruption on the input video or images; shot noise is caused by the discrete nature of photons. Some works [10][11][13][18] have shown that common corruptions can degrade the deep neural networks performance significantly.

Hendrycks et al. [18] firstly proposed benchmarks ImageNet-C and CIFAR10-C to measure the corruption robustness of models against these corruptions in the test dataset. In the following, the corruption robustness benchmarks are proposed for object detection [32], semantic segmentation [21] and pose estimation [44]. This image-based corruption robustness uses the same 15 types of corruption for evaluation. R.Taori et al. [41] empirically show that the robustness on synthetic common corruption dataset does not imply robustness on other distribution shifts arising in nature. One of

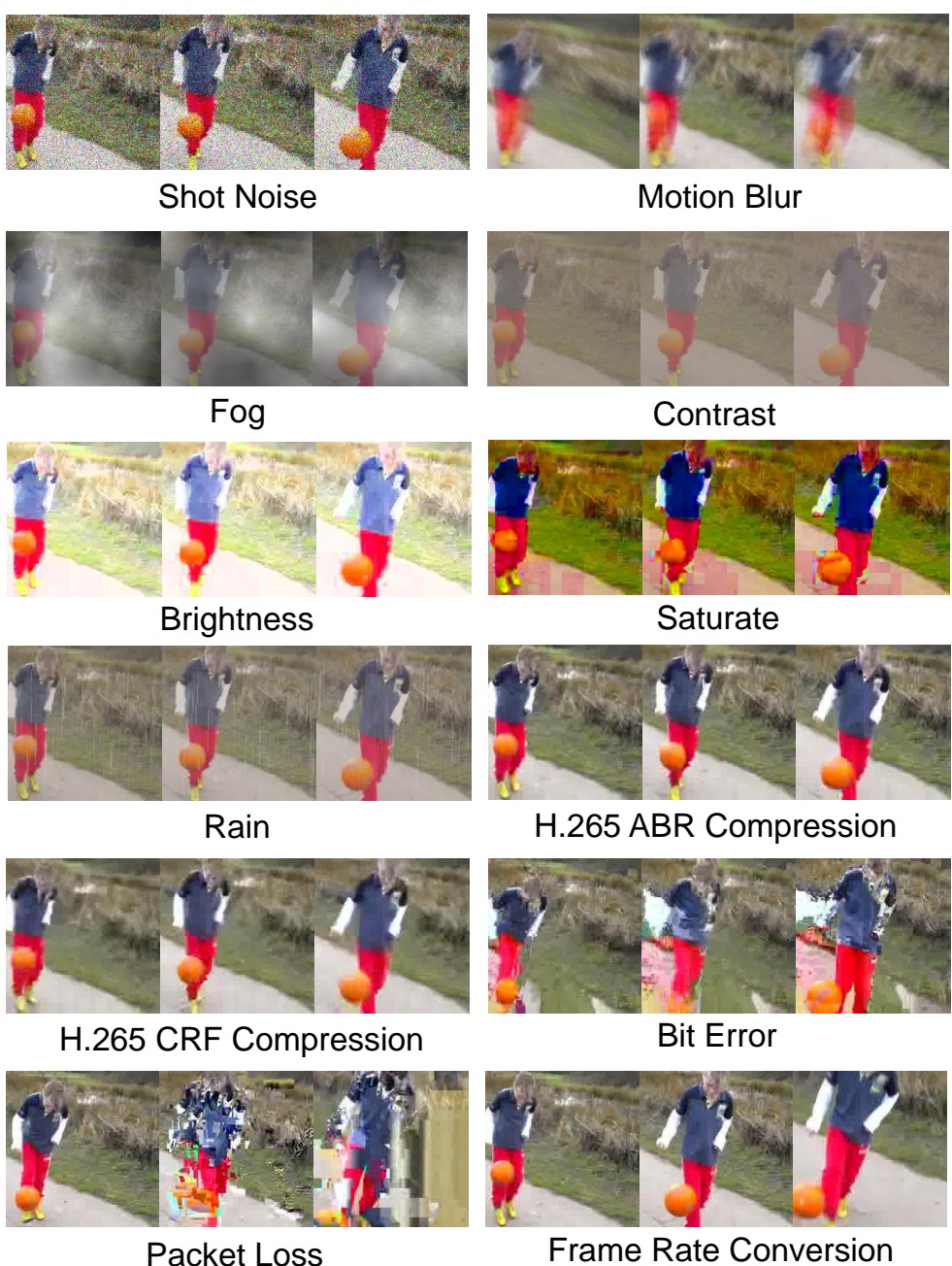

Figure 1: Our proposed corruption robustness benchmark consists of 12 types of corruptions with five levels of severity for each video. In the examples above, we use uniform sampling to extract 3 frames from each corrupted video, the sampling interval is 10 frames. (More corrupted samples can be found in the supplementary material)

the natural distribution shifts [15][37] uses video as input while it is to measure the robustness of the image classification model to small changes in continuous video frames. However, Hendrycks et al. [17] argue that the robustness intervention on ImageNet-C improves the robustness to real-world blurry images. Taken together, these findings indicate that evaluation on multiple distribution shifts leads to a more comprehensive understanding of the robustness of models. In this work, we extend the distribution shifts to temporal corruptions for video-based classifications. We use 12 types of common corruptions as a systematic proxy of real-world corruptions to understand the behavior of

models in the wild. We are the first one to conduct an extensive analysis of the corruption robustness of deep neural networks in video classification.

## 2.2 Video Classification

Video classification has been one of the active research areas in computer vision. Recently, with the great progress of deep learning techniques, various deep learning architectures have been proposed. Most famous deep learning approaches for action recognition are based on two stream model [38]. The Two-stream 2D CNN framework first introduced by Simonyan and Zisserman [38], employs a two-stream CNN model consisting of a spatial network and a temporal network. Wang et al. [45] divided each video into three segments and processed each segment with a two-stream network. Fan et al. [7] presented an efficient and memory-friendly video architecture to capture temporal dependencies across frames.

Plenty of works have extended the success of 2D models in image classification to recognize action in videos. Tran et al. [42] proposed a 3D CNN based on VGG models, named C3D, to learn spatio-temporal features from video sequences. Carreira and Zisserman [4] proposed converting 2D ConvNet [39] to 3D ConvNet by inflating the filters and pooling kernels with an additional temporal dimension. Xie et al. [47] replaced many of the 3D convolutions in I3D [4] with low-cost 2D convolutions to seek a balance between speed and accuracy. Tran et al. [43] show that decomposing the 3D convolutional filters into separate spatial and temporal components yields significantly gains in accuracy. Feichtenhofer et al. [9] designed a two-stream 3D CNN framework that operates on RGB frames at low and high frame rates to capture semantic and motion, respectively. More recently, Feichtenhofer et al. [8] presented X3D as a family of efficient video networks. As the great success of Transformers-based models achieved in image classification [6], several premier works [2][28] have adopted Transformer-based architectures for video-based classification. Gedas et al. [2] first proposed applying several scalable self-attention designs over space-time dimension and suggested the "divided attention" architecture, named TimeSformer. In this paper, we establish the benchmark and extensive experiments to evaluate the robustness of these 2D CNN and 3D CNN methods, as well as the transformer-based methods.

# 3 Benchmark Creation

## 3.1 Corruption Robustness in Classification

When a computer vision system is deployed in the real world, the system may encounter different types of common corruptions caused by unforeseeable environmental changes (e.g., rain, fog) or system errors (e.g., network error). The sustainability of models under corruption is defined as corruption robustness [18]. It measures the average-case performance of models. To be more specific, the first line of corruption robustness considers the cases where the corruptions in the test data are unseen in training. All the corruptions follow a distribution $P_C$ in the real world: $c \sim P_C$. For each type of corruption $c$, the corrupted samples $(c(x), y)$ follow the distribution $P_{(c(X),Y)}$: $(c(x), y) \sim P_{(c(X),Y)}$. For a classifier $f : X \rightarrow Y$, a general form of corruption robustness is given by

$$R_f := \mathbb{E}_{c \sim P_C}[\mathbb{E}_{(c(x),y) \sim P_{(c(X),Y)}}[f(c(x)) = y)]], \tag{1}$$

where $x$ is the input, and $y$ is the corresponding target label.

## 3.2 Metric Explanation

While it is impossible to evaluate the expectation in Equation 1 exactly by considering all corrupted data, we can estimate it using finite types of corruptions and limited corrupted samples. With the estimation, we use $PC$ to evaluate the robustness of models against single corruption in classification:

$$PC_c = \frac{1}{N_s} \sum_{s=1}^{N_s} CA_{c,s}, \tag{2}$$

where $CA_{c,s}$ is the classification accuracy on dataset samples with corruption type $c$ under severity level $s$. Then we estimate the model corruption robustness with mean performance under corruption $mPC = \frac{1}{N_c} \sum_{c=1}^{N_c} PC_c$; $N_c$ and $N_s$ are the number of corruption types and the number of severity levels. Hence, the total number of corruptions applied on clean samples is $N_c \times N_s$. It follows the similar metric as [32]. Because the classification accuracy of models in video classification tasks highly relies on input lengths, frame rates, sampling methods, and architectures [5], the absolute classification accuracy cannot depict the robustness of models fairly under different settings. To evaluate the robustness of models based on their generalization ability on clean data, relative mean performance under corruption is also used in this paper, which is given as

$$rPC = \frac{mPC}{CA_{clean}}, \tag{3}$$

where $CA_{clean}$ is the classification accuracy of models on video classification datasets without any corruption. For our corruption robustness benchmark, we use mPC to rank the approaches as shown in our Github repository [1]. The rPC indicates the relative performance of models under corruptions, and it helps disentangle the gain from clean performance and the gain from generalization performance to corruptions. We use it to provide additional insight into the robustness of models.

### 3.3 Benchmark Datasets

The robust video classification benchmark contains two benchmarks datasets: Mini Kinetics-C and Mini SSV2-C. In each dataset, we apply $N_c = 12$ types of corruptions with $N_s = 5$ levels of severity on clean data to estimate the possible unseen test data. Most of the previous image-related benchmarks [18][32][44] use the same 15 types of corruptions to benchmark the general robustness without proposing new types of corruptions. Different from image-based tasks, video classification tasks use video as the inputs. From the aspect of video processing and formation, corruptions can generate at various stages. We summarize the corruptions adopted for the proposed benchmark below.

**Video Acquisition:** *Shot Noise, Motion Blur, Fog, Contrast, Brightness, Saturate, Rain.*
**Video Processing:** *H.265 ABR Compression, H.265 CRF Compression, Bit Error, Packet Loss, Frame Rate Conversion.*

To be specific, shot noise [1][31], motion blur [20][23], fog [22][48], contrast, brightness, and saturate corruptions widely exist in images and videos captured in the real world. Hence, they are also presented in image-based corruption robustness benchmarks. Besides, we add rain corruption [26][27] in our benchmark because it is the most common bad weather in nature. The other five types of corruptions are generated in the pipeline of **video processing** [35][46]. We firstly introduce two types of corruption caused by video compression. In many video-based applications with the communication network, the raw video requires high bandwidth, but the bandwidth of network is limited. As a result, compression is compulsory for these real-world applications. The first type of compression, H.265 ABR compression, uses the popular codec H.265 for compression. The compression targets an Average Bit Rate; it is a lossy video compression that generates compression artifacts. Another type of corruption is caused by H.265 CRF Compression. Different from Average Bit Rate, Constant Rate Factor (CRF) is another encoding mode by controlling quantization parameter. It introduces compression artifacts as well. Bit Error and Packet Loss corruptions come from video transmission due to imperfect channels in the real world [25]. The error will propagate in the subsequent channel and cause more severe corruptions. Frame Rate Conversion means the frame rate of test data can differ from the frame rate of training data. Because of limited bandwidth, the video capturing system deployed in the wild may use a lower frame rate for transmission. Specifically, we categorize motion blur, ABR compression, CRF compression, bit error, packet loss, and frame rate conversion into temporal corruptions. These corruptions are generated based on a set of continuous frames. The rest of corruptions are categorized into spatial corruptions because they only depend on the content in a single frame. We show some corruption examples in Figure 1.

**Mini Kinetics-C:** Kinetics [4] are the most popular benchmark for video classification tasks in the past few years. It contains 240K training videos and 20K validation videos of 400 classes. Each video in Kinetics lasts for 6-10 seconds. Considering the scale of Kinetics, it is more practical to use half of the dataset for training and evaluation. Moreover, it enables us to conduct experiments on more

---

[1]`https://github.com/Newbeeyoung/Video-Corruption-Robustness`

settings with less infusion for video classification tasks. To create Mini Kinetics-C, We randomly pick 200 classes from Kinetics to create Mini Kinetics. Then we apply 12 types of corruptions with 5 levels of severity on the validation dataset of Mini Kinetics, so Mini Kinetics-C is 12x5 times the validation dataset in Mini Kinetics.

**Mini SSV2-C:** SSV2 dataset [14] consists of 168K training videos and 24K validation videos of 174 classes. Each video lasts for 3-5s. Different from Kinetics, the SSV2 dataset highly relies on temporal information for classification. It also has less background information. Similar to Mini Kinetics-C, we randomly choose 87 classes from the original SSV2 and apply the corruptions on the validation dataset to create Mini SSV2-C.

## 4 Benchmark Study

With the proposed corrupted video classification datasets, we raise several obvious questions: **Q1:** how robust are current models? **Q2:** how robust are the models against spatial and temporal corruptions? **Q3:** Are the trends of increasing video classification model generalization ability and efficiency aligned with the target of improving their robustness?

We conduct a large-scale analysis involving more than 60 video classification models and the proposed two datasets. Each dataset contains 60 different corruptions. Based on the study, we answer the three questions in Section 4.2, 4.3 and 4.4 respectively. Due to the space limit, we include ablation studies of the proposed benchmark in the supplementary material.

### 4.1 Training and Evaluation

We follow the protocol in [5] and use 32-frame input for training and evaluation. Because the batch size has a large impact on the performance of models, we use a batch size of 32 to train most models. For Transformer-based model, we slightly reduce the batch size to 16 due to GPU capacity limits. For data preprocessing and augmentation, we extract the image frame from videos and resize the size smaller to 240. We then apply multi-scale crop augmentation on each clip input and resize them to 224x224. We train all the models with an initial learning rate of 0.01.

In video classification, the model can be evaluated at clip level and video level settings using uniform sampling and dense sampling. For our evaluation, we randomly choose one frame from the first segment of video and extract frames with fixed stride as uniform sampling, at the clip level. This setting guarantees the efficiency of evaluation without sacrificing much accuracy. The comparison with other settings is shown in the supplementary material.

### 4.2 Q1: Benchmarking Robustness of SOTA Approaches

In our benchmark study, we train the CNN-based and the transformer-based models with clean data and evaluate them on the corrupted data. It is a standard-setting under the robust generalization study [12, 18], which assumes that the model is not able to know the exact problem in the deployment in advance. Therefore, we evaluate each model on 12 types of corrupted videos and average their performance. The evaluation explores the impact of every single corruption on the model. Besides, the average performance of the model on corruptions indicates the average-case robustness of the model against unseen corruptions in nature.

We trained several networks, including S3D, I3D, 3D ResNet, SlowFast, X3D, TAM, and TimeSformer on clean Mini Kinetics and Mini SSV2. S3D, I3D, and 3D ResNet are the previous SOTA methods for video classification, while SlowFast, X3D, and TAM claim to achieve the SOTA performance with less computational cost. We use ResNet50 as the backbone of SlowFast, X3D, and TAM for a fair comparison. Different from previous models, TimeSformer is transformer-based, and it achieves competitive performance against the most advanced CNN-based models. During training, we used uniform sampling to extract 32 frames as input. We then evaluated the models using uniform sampling at 32 frames at the clip level. The corruption robustness of the state-of-the-art spatial-temporal models is shown in Table 1. It shows that TimeSformer achieves the best performance on clean accuracy, mPC, and rPC on Mini Kinetics-C. Although TAM performs the best on clean Mini SSV2, its mPC and rPC are 4.0% and 8.2% respectively, lower than TimeSformer. It shows that the transformer-based model outperforms the CNNs in terms of corruption robustness.

Table 1: Corruption robustness of architectures on the corrupted Mini Kinetics and Mini SSV2. All the models are trained with 32-frame inputs and evaluated at clip level. I3D and S3D use a backbone of InceptionV1. X3D-M, TAM and SlowFast use a backbone of ResNet50.

| Approach | Clean | mPC | rPC | Spatial | | | | | | Temporal | | | | | |
|---|---|---|---|---|---|---|---|---|---|---|---|---|---|---|---|
| | | | | Shot | Rain | Fog | Contrast | Brightness | Saturate | Motion | Frame Rate | ABR | CRF | Bit Error | Packet Loss |
| Mini Kinetics-C | | | | | | | | | | | | | | | |
| S3D [47] | 69.4 | 56.9 | 82.0 | 50.8 | 51.5 | 47.6 | 46.4 | 62 | 56.8 | 54.9 | 68.3 | 62.8 | 59.1 | 59.9 | 62.9 |
| I3D [4] | 70.5 | 57.7 | 81.8 | 56.1 | 50.5 | 45.8 | 46.3 | 63.1 | 57.5 | 56.5 | 68.9 | 64.3 | 60.8 | 59.4 | 62.9 |
| 3D ResNet-18 [16] | 66.2 | 53.3 | 80.5 | 47.7 | 45.8 | 40.5 | 43 | 58.9 | 53.5 | 53.4 | 65.1 | 60.1 | 56.4 | 55.8 | 59.3 |
| 3D ResNet-50 [16] | 73.0 | 59.2 | 81.1 | 49.6 | 47.6 | 49.1 | 51.5 | 65.8 | 60.0 | 59.3 | 71.9 | 65.9 | 61.6 | 62.2 | 65.9 |
| SlowFast 8x4 [9] | 69.2 | 54.3 | 78.5 | 33.4 | 51.7 | 40.7 | 46.5 | 61.2 | 54.1 | 54.9 | 68.5 | 63.1 | 59.1 | 56.7 | 62.2 |
| X3D-M [8] | 62.6 | 48.6 | 77.6 | 33.2 | 44.4 | 36.9 | 40.9 | 54.8 | 48.6 | 47.3 | 61.0 | 55.9 | 53.6 | 51.3 | 55.9 |
| TAM [7] | 66.9 | 50.8 | 75.9 | 47.6 | 47.8 | 35.8 | 38.2 | 58.0 | 45.3 | 44.5 | 65.0 | 58.1 | 54.3 | 55.9 | 59.7 |
| TimeSformer [2] | 82.2 | 71.4 | 86.9 | 74.7 | 70.5 | 55.3 | 62.7 | 76.1 | 69.6 | 70.8 | 81.1 | 76.5 | 73.0 | 71.5 | 75.4 |
| Mini SSV2-C | | | | | | | | | | | | | | | |
| S3D [47] | 58.2 | 47.0 | 81.8 | 40.3 | 27.5 | 43.3 | 48.1 | 52.3 | 47.7 | 36.1 | 47.5 | 53.2 | 52.4 | 43.8 | 50.8 |
| I3D [4] | 58.5 | 47.8 | 81.7 | 43.7 | 30.5 | 43.4 | 46.3 | 53.1 | 49.5 | 36.3 | 49.0 | 53.7 | 52.6 | 44.3 | 51.3 |
| 3D ResNet-18 [16] | 53.0 | 42.6 | 80.3 | 34.1 | 21.9 | 38.0 | 42.9 | 48.0 | 42.5 | 34.9 | 42.9 | 49.1 | 47.8 | 40.3 | 46.9 |
| 3D ResNet-50 [16] | 57.4 | 46.6 | 81.2 | 39.8 | 24.4 | 39.7 | 47.8 | 51.5 | 45.1 | 36.1 | 47.3 | 52.3 | 50.5 | 43.7 | 50.5 |
| SlowFast 8x4 [9] | 48.7 | 38.4 | 78.8 | 26.9 | 34.0 | 32.4 | 34.9 | 40.4 | 34.6 | 27.7 | 37.1 | 44.4 | 44.0 | 36.1 | 41.8 |
| X3D-M [8] | 49.9 | 40.7 | 81.6 | 32.9 | 37.0 | 36.6 | 39.4 | 44.2 | 38.8 | 28.5 | 39.5 | 46.0 | 45.9 | 38.4 | 43.9 |
| TAM [7] | 61.8 | 45.7 | 73.9 | 39.1 | 19.2 | 43.3 | 52.1 | 53.7 | 45.0 | 33.4 | 49.9 | 55.5 | 54.5 | 47.9 | 54.3 |
| TimeSformer [2] | 60.5 | 49.7 | 82.1 | 41.9 | 52.2 | 41.4 | 47.1 | 54.3 | 50.0 | 45.1 | 57.1 | 54.9 | 53.8 | 46.7 | 52.0 |

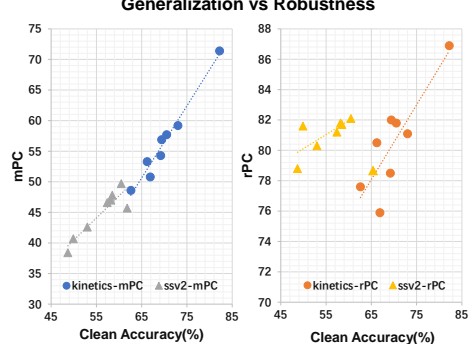

Figure 2: The generalization and robustness of the SOTA approaches shown in Table 1.

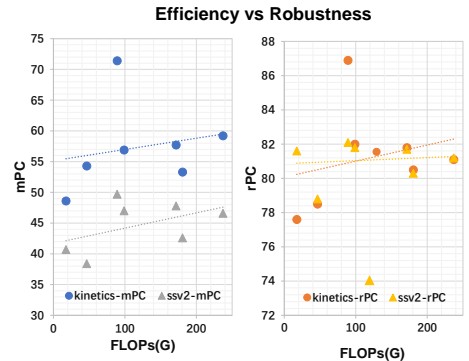

Figure 3: The efficiency and robustness of the SOTA approaches shown in Table 1.

## 4.3 Q2: Robustness w.r.t Spatial and Temporal Corruptions

**Robustness w.r.t spatial corruptions.** As we can observe from Table 1, models trained on Mini Kinetics are vulnerable to fog, contrast, and shot noise, while models trained on Mini SSV2 are vulnerable to rain, shot noise, and fog. We hypothesize that rain corruption deteriorates temporal information more than contrast, so it has a larger impact on the SSV2 dataset. To study the robustness of models against spatial and temporal corruptions, we present the mPC and rPC separately in Table 2.

Table 2: mPC and rPC on spatial and temporal corruptions

| Approach | Mini Kinetics-C | | | | Mini SSV2-C | | | |
|---|---|---|---|---|---|---|---|---|
| | spatial | | temporal | | spatial | | temporal | |
| | mPC | rPC | mPC | rPC | mPC | rPC | mPC | rPC |
| S3D [47] | 52.5 | 75.7 | 61.3 | 88.4 | 43.2 | 74.2 | 47.3 | 81.2 |
| I3D [4] | 53.2 | 75.5 | 62.1 | 88.1 | 44.4 | 75.9 | 47.9 | 81.8 |
| 3D ResNet-18 [16] | 48.2 | 72.9 | 58.3 | 88.1 | 37.9 | 71.5 | 43.7 | 82.4 |
| 3D ResNet-50 [16] | 53.9 | 73.9 | 64.5 | 88.3 | 41.4 | 72.1 | 46.7 | 81.4 |
| SlowFast 8x4 [9] | 47.9 | 69.3 | 60.7 | 87.8 | 33.9 | 69.5 | 38.5 | 79.1 |
| X3D-M [8] | 43.1 | 68.9 | 54.1 | 86.5 | 38.2 | 76.5 | 40.4 | 80.9 |
| TAM [7] | 45.5 | 67.9 | 56.2 | 84.0 | 42.1 | 68.1 | 49.3 | 79.7 |
| TimeSfomer [2] | 68.2 | 83.0 | 74.7 | 90.9 | 47.8 | 79.0 | 51.6 | 85.2 |

It shows that the CNN-based models have similar rPC on spatial corruptions on both datasets, ranging from $67.9 \sim 76.5\%$. TimeSformer achieves the highest performance on all metrics on both datasets.

**Robustness w.r.t temporal corruptions.** A trend can be observed that the robustness of models against temporal corruptions increases with the clean accuracy from Table 1. Table 2 also shows that the models with the best generalization have the highest mPC on temporal corruptions. Models trained on Mini Kinetics handle all types of temporal corruptions well. Interestingly, although Bit Error and Packet Loss can propagate and augment corruptions in the consequent frames, the performance of models for Mini Kinetics degrades less on these two corruptions, even if some frames are not recognizable by human. However, the models for Mini SSV2 are more sensitive to temporal corruptions. Especially, the accuracy drops by $15.4 \sim 28.4\%$, $11.5 \sim 14.4\%$ and $3.4 \sim 11.9\%$ on motion blur, bit error, and frame rate conversion for Mini SSV2-C. It only drops by $11.4 \sim 22.4\%, 10.7 \sim 12.5\%$ and $1.1 \sim 1.6\%$ for Mini Kinetics-C, respectively. We also find that rPC of models on temporal corruptions for Mini SSV2 is $4 \sim 8\%$ lower than them for Mini Kinetics in Table 2.

### 4.4 Q3: Trade off between Robustness, Generalization and Efficiency

As seen in Figure 2, when the clean accuracy of models improves, the mPC of models improves as well. On both datasets, we observe the same trend that rPC of models improves slightly when the clean accuracy improves. On Mini Kinetics-C, the clean accuracy, mPC, and rPC of 3D ResNet50 are $10.4\%, 10.6\%$, and $13.5\%$ higher than those of X3D-M, respectively. The results show that models with better generalization are also more robust against the common corruptions.

From the dimension of computation cost in Figure 3, there is a clear trend that the mPC increases with floating-point operations (FLOPs) per sample. X3D-M, SlowFast and 3D ResNet-50 use the same backbone of ResNet50, where X3D-M and SlowFast reduce the FLOPs with different techniques. However, on both datasets, the rPC decreases from $81\%$ to $78\%$ when the FLOPs of 3D ResNet-50 reduce from 180.2G to 46.5G FLOPs of SlowFast network. Similarly, the rPC of X3D-M decreases to $77.6\%$ on Mini Kinetics-C though its FLOPs reduces to 17.5G. TAM is an outlier in terms of rPC, as it is the only model using pure 2D CNN. Moreover, the claimed efficient approaches SlowFast, X3D-M and TAM have relatively lower rPC on temporal corruptions in Table 2. It appears that there is a trade-off between efficiency and corruption robustness.

We also explore the impact of capacity on the robustness of models. For the CNN-based models, when we increase the backbone size from ResNet18 to ResNet50 for the 3D ResNet approach, the improvements of clean accuracy, mPC and rPC on both datasets indicate that the robustness increases with model capacity. Previous robustness benchmark studies [17] [44] in image domain also obtain the same finding. Besides, TimeSformer has a much larger capacity of 121.3M parameters, which is around 3 times of the largest CNN-based model 3D ResNet-50 with 46.4M parameters.

## 5 Training Models on Corrupted Videos

In our benchmark study, we train the model with clean data and evaluate them on the corrupted data to examine the average-case robustness of the model against unseen corruptions. However, it is common to train the model with noise to increase the robustness in machine learning problem [3, 24]. We therefore train the model with random Gaussian noise in the video to understand the impact. We also extend Gaussian noise to the proposed corruptions in our benchmark. In the real-world deployment, it is possible to estimate at least one type of corruption under the deployment circumstance, but we cannot foresee all types of corruption encountered. Intuitively, we can train the model with a certain type of corruption to improve evaluation accuracy on the corrupted videos under a supervised learning setting. In robust generalization, it becomes significant to understand whether the robustness obtained from training with one type of corruption can transfer to other types of corruptions [12].

### 5.1 Training with Gaussian noise

Gaussian data augmentation has been widely used to improve the robustness of models in image-related tasks [13, 24, 29, 33]. To examine the impact of it on corruption robustness, we train models with additive Gaussian noise on clean data for both datasets. We use the standard deviation of 0.1 and 0.2. In Table 3, we show that the clean accuracy, mPC of 3D ResNet-18 trained with Gaussian data augmentation are lower than models trained on clean data. It contradicts the results for

image corruption robustness, where the average-case corruption robustness is improved when the Gaussian data augmentation is applied. More specifically, Gaussian data augmentation degrades the performance of models on 10 out of 12 types of corruption. We conjecture that corrupting each frame with random Gaussian noise introduces extra temporal information into the video, deteriorating the original temporal information besides the spatial information. It only improves the robustness against shot noise and rain, because they are semantically similar. The results show that vanilla Gaussian data augmentation is not able to generalize to most types of unseen corruptions, which makes robustness improvement an open question from the aspect of data augmentation.

Table 3: Corruption robustness of 3D ResNet-18 with/without Gaussian data augmentation on the corrupted Mini Kinetics and Mini SSV2. The std indicates the standard deviation of Gaussian noise.

| Approach | Clean | mPC | rPC | Spatial | | | | | | Temporal | | | | | |
| | | | | Shot | Rain | Fog | Contrast | Brightness | Saturate | Motion | Frame Rate | ABR | CRF | Bit Error | Packet Loss |
|---|---|---|---|---|---|---|---|---|---|---|---|---|---|---|---|
| | | | | | | | Mini Kinetics-C | | | | | | | | |
| 3D ResNet-18 | **66.2** | **53.3** | **80.5** | 47.7 | 45.8 | **40.5** | **43.0** | **58.9** | **53.5** | **53.4** | **65.1** | **60.1** | **56.4** | **55.8** | **59.3** |
| 3D ResNet-18(std=0.1) | 63.6 | 50.1 | 78.7 | **61.9** | **50.6** | 28.7 | 28.2 | 56.4 | 48.1 | 45.4 | 62.0 | 56.8 | 52.6 | 53.5 | 56.4 |
| 3D ResNet-18(std=0.2) | 57.8 | 43.8 | 75.8 | 60.7 | 46.6 | 14.0 | 16.9 | 51.2 | 41.4 | 37.2 | 56.4 | 51.4 | 48.4 | 49.3 | 51.9 |
| | | | | | | | Mini SSV2-C | | | | | | | | |
| 3D ResNet-18 | **53.0** | **40.8** | 77.0 | 34.1 | 21.9 | **38.0** | **42.9** | **48.0** | **42.5** | **34.9** | **42.9** | **49.1** | **47.8** | **40.3** | **46.9** |
| 3D ResNet-18(std=0.1) | 47.5 | 36.0 | 75.7 | 43.9 | 28.5 | 25.5 | 26.0 | 43.5 | 40.2 | 25.4 | 36.6 | 43.8 | 43.2 | 34.5 | 40.4 |
| 3D ResNet-18(std=0.2) | 45.7 | 36.2 | **79.2** | **46.4** | **36.5** | 25.1 | 25.5 | 42.8 | 39.9 | 26.1 | 34.7 | 42.4 | 41.7 | 33.7 | 39.0 |

## 5.2 Training with Proposed Corruptions

Beyond evaluating the models trained on clean and Gaussian noise corrupted data with our proposed benchmark, we also directly train the network with our proposed corruptions. We use the backbone of 3D ResNet-18. Similar to [12], we use a subset of the standard Kinetics to train the network. It contains 40 classes in the mini kinetics dataset. For each video in the dataset, we apply a single type of corruption on the clean data for training, where the corruption has a severity level of 3 as defined in our benchmark.

The results in Table 4 show that most of the models achieve the best performance on the corruption they are trained on. However, our proposed benchmark aims to evaluate the robust generalization ability of models under the condition that we may not know the corruption exactly in advance. Hence, we test the models on other types of corruption as well. We find that the models do not generalize to other types of corruption well. 9 out of 12 models trained on single corruption obtain lower mPC than the model trained on clean data. Additionally, training models with corruption directly degrades the performance on clean data. 10 out of 12 models trained on single corruption have lower clean accuracy.

Table 4: Corruption robustness of 3D ResNet-18 on the 40-class corrupted Mini Kinetics.

| Model | Clean | mPC | rPC | Spatial | | | | | | Temporal | | | | | |
| | | | | Shot | Rain | Fog | Contrast | Brightness | Saturate | Motion | Frame Rate | ABR | CRF | Bit Error | Packet Loss |
|---|---|---|---|---|---|---|---|---|---|---|---|---|---|---|---|
| Shot Noise | 71.5 | 61.4 | 85.9 | **73.6** | 69.2 | 34.8 | 44.2 | 61.2 | 57.0 | 65.0 | 70.1 | 68.3 | 65.6 | 62.2 | 65.2 |
| Rain | 73.7 | 63.1 | 85.6 | 57.6 | **74.9** | 36.8 | 55.3 | 66.5 | 59.9 | 68.2 | 72.3 | 69.6 | 67.0 | 63.1 | 66.4 |
| Fog | 41.6 | 41.8 | **100.5** | 34.3 | 46.3 | **55.7** | 52.5 | 34.8 | 28.2 | 49.8 | 46.4 | 42.1 | 39.5 | 35.3 | 36.6 |
| Contrast | 49.6 | 47.5 | 95.8 | 36.2 | 56.8 | 38.9 | **67.1** | 46.6 | 38.0 | 55.5 | 51.8 | 48.7 | 46.1 | 41.5 | 42.8 |
| Brightness | 71.2 | 59.8 | 84.0 | 51.4 | 63.2 | 33.2 | 48.7 | **74.3** | 60.7 | 62.9 | 68.6 | 66.8 | 64.0 | 60.4 | 62.9 |
| Saturate | 75.4 | 61.9 | 82.1 | 56.8 | 66.0 | 28.3 | 47.6 | 69.3 | **68.5** | 67.6 | 72.7 | 69.6 | 65.7 | 63.7 | 67.3 |
| Motion Blur | 71.3 | 60.6 | 85.0 | 50.9 | 61.4 | 39.7 | 53.5 | 61.9 | 57.4 | **75.7** | 70.4 | 68.2 | 65.8 | 60.0 | 62.6 |
| Frame Rate | **80.3** | 65.5 | 81.6 | 53.1 | 65.9 | 41.2 | 54.4 | 70.1 | 65.3 | 71.6 | 77.2 | 75.4 | 71.4 | 68.5 | 71.8 |
| ABR | 79.4 | **66.6** | 83.9 | 56.4 | 68.2 | 44.1 | 57.2 | 69.5 | 65.7 | 72.3 | **77.3** | 75.9 | 73.4 | 67.9 | 71.8 |
| CRF | 78.8 | 65.8 | 83.5 | 55.4 | 67.3 | 40.1 | 53.2 | 67.5 | 65.8 | 72.0 | 76.8 | **76.8** | 74.4 | 68.4 | 71.5 |
| Bit Error | 77.8 | 66.5 | 85.5 | 55.1 | 68.2 | 42.0 | 57.0 | 70.9 | 65.3 | 68.5 | 76.6 | 74.4 | 70.9 | **73.4** | **75.1** |
| Packet Loss | 77.3 | 64.6 | 83.6 | 56.5 | 67.4 | 36.9 | 52.0 | 69.9 | 64.6 | 67.4 | 74.7 | 72.0 | 68.5 | 71.7 | 73.0 |
| Clean | 78.9 | 65.8 | 83.4 | 57.4 | 70.0 | 39.0 | 54.0 | 70.0 | 66.5 | 71.8 | 76.8 | 74.2 | 70.4 | 67.3 | 72.0 |

# 6 Conclusion

In this paper, we have proposed a robust video classification benchmark (Mini Kinetics-C and Mini SSV2-C). It contains various types of temporal corruptions, which distinguish themselves from spatial

level image corruptions in all image robustness benchmarks. With the proposed benchmarks, we have conducted an exhaustive large-scale evaluation on the corruption robustness of the state-of-the-art CNN-based and transformer-based models in video classification. Based on the evaluation results, we have drawn several conclusions in terms of robust architecture and training. We also provide some insights into the impact of temporal information on model robustness, which has been largely unexplored in the existing research works. As a new dimension of research in video classification, robustness is to be systematically studied in future works due to its universality and significance.

## Acknowledgments and Disclosure of Funding

This work was done at EEE Rapid-Rich Object Search (ROSE) Lab, Nanyang Technological University. This research is supported by the NTU-PKU Joint Research Institute (a collaboration between the Nanyang Technological University and Peking University that is sponsored by a donation from the Ng Teng Fong Charitable Foundation).

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
