# OpenReview forum: "Benchmarking the Robustness of Spatial-Temporal Models Against Corruptions"
_NeurIPS.cc/2021/Track/Datasets_and_Benchmarks/Round2 — NeurIPS 2021 Datasets and Benchmarks Track (Round 2)_

### Official Review · Reviewer_nWFA · 2021-09-16
**A useful benchmark to test robustness of video classification models against data corruption**

**Rating:** 6
**Confidence:** 3
**Clarity:** The paper is well written.

**Strengths:**

Studying the robustness of models against corruptions that may occur in real life applications is important both from a scientific and a practical point of view. This work does that in the context of video classification and is a natural next step following similar benchmarks and studies in the context of image classification. I think that the benchmark that the authors created could be useful to the computer vision community in the pursuit of developing accurate and robust models.

**Weaknesses:**

I think that the main contribution of this work is the benchmark itself. I would therefore expect some more details such as: how were the corruptions implemented? how did you determine the severity scale? What are the different severity levels for compression?
I couldn't find any of that, not even in the supplementary material, expect for a referral to the code in Github.

I think that from the results presented in the paper, it is difficult to draw a clear conclusion regarding the effect of model capacity/computation on robustness. The fact that performance on clean data is correlated with mPC and both are correlated with model capacity is not surprising. We know that by and large more capacity leads to better generalization. The more interesting metric to look at, in my opinion, is rPC vs. capacity/computation. With respect to that, Figure 3 does not show a significant correlation. Furthermore, while there is a drop in rPC between 3D ResNet-50 to 3D ResNet-18, it is not significant. I think that more direct comparisons of that sort, between similar models with backbones of different sizes are necessary. Also, it would be interesting to include transformer-based models in such an experiment. Currently only one such model is considered.

**Additional Feedback:**



**Correctness:**

The claims seem to be correct. I can't say much about the benchmark construction because the authors did not provide sufficient details.

**Documentation:**

The authors do not provide sufficient details about the benchmark construction.

**Ethics:**

I don't see any ethical concern.

**Relation To Prior Work:**

Prior work is well covered.

**Summary And Contributions:**

This paper presents a benchmark for testing the robustness of action recognition models in videos against different types of data corruptions which may occur in real life. In addition to corruptions which are spatial in nature and were previously considered for images, the paper introduces several types of temporally-related corruptions which may occur in the process of capturing and transmitting videos. Using the benchmark, the authors conduct a study on the relations between robustness, generalization, and models' type and efficiency.

---

> ### Author Response · Authors · 2021-09-29
> **Author Response**
>
> **Q: I think that the main contribution of this work is the benchmark itself. I would therefore expect some more details such as: how were the corruptions implemented? how did you determine the severity scale? What are the different severity levels for compression? I couldn't find any of that, not even in the supplementary material, expect for a referral to the code in Github.**
>
> **A:**  Thanks for your encouraging comments! In the original Github repository, we have provided code examples ‘create_mini_kinetics_c.py’ and ‘create_mini_ssv2_c.py’ for researchers to construct our benchmark. All the implementation details, including the severity of corruption and level of compression, are presented in the code.
>
> To further clarify the implementation of the benchmark, we added Section 3 and Table 1 to show the implementation detail of each corruption in the supplementary.
>
> **Q: I think that from the results presented in the paper, it is difficult to draw a clear conclusion regarding the effect of model capacity/computation on robustness. The more interesting metric to look at, in my opinion, is rPC vs. capacity/computation. With respect to that, Figure 3 does not show a significant correlation. Furthermore, while there is a drop in rPC between 3D ResNet-50 to 3D ResNet-18, it is not significant. I think that more direct comparisons of that sort, between similar models with backbones of different sizes are necessary.**
>
> **A:** Thanks for your valuable suggestion. We have modified our discussion in Section 4.4 to emphasize the trend of rPC vs computation and capacity. From the computation aspect, X3D-M, SlowFast, and 3D ResNet-50 use the same backbone, while the previous two methods reduce the computational cost with some techniques. However, the rPC of claimed efficient methods drops when compared with 3D ResNet-50.
>
> Besides, we revised the visualization of rPC vs FLOPs (efficiency) in Figure 3 to make the trend more obvious. When we discard TAM (2D CNN-based) and TimeSformer(transformer-based), the rest of the models use a similar backbone of 3D CNN. The trend is more significant and we have visualized it in Section 9 in the Supplementary.
>
> **Q: Also, it would be interesting to include transformer-based models in such an experiment. Currently only one such model is considered.**
>
> **A:** Currently, we do not have enough resources to include more transformer-based models in our benchmark due to the limitations of time and computation capacity. We will update the performance of more transformer-based models in our Github repository in the future. We also welcome the submission of transformer-based model results on our benchmark from other researchers.

---

### Official Review · Reviewer_MVT9 · 2021-09-19
**The authors propose the first benchmark for assessing the robustness of action recognition models against real-world spatial and temporal corruptions**

**Rating:** 7
**Confidence:** 3

**Strengths:**

To my knowledge, while many prior works have assessed the robustness of 2D image classification networks with respect to common spatial artifacts, none have explored the effect of artifacts on video classification / action recognition. This benchmark will be the first, which will further understanding about the effect that both spatial and temporal artifacts have on action recognition tasks.

**Weaknesses:**

The authors do not provide model definitions, nor the code used to actually evaluate each model on the Mini Kinetics-C and SSV-C datasets. This limits the reproducibility of the benchmark results in the paper, and the leaderboard found in the Github repository.

**Additional Feedback:**

•	In Section 2.2, I recommend including another work by D Tran, et al. in which video classification networks were further improved by decomposing 3D spatiotemporal convolutions into 2D spatial and 1D temporal convs with an activation between (R(2+1)D networks): D Tran, et al. “A Close Look at Spatiotemporal Convolutions for Action Recognition” DOI:10.1109/CVPR.2018.00675
-	How is the frame rate conversion performed exactly? Is the video data re-interpolated? What interpolation scheme is used? Please clarify this in the text.
-	How did the authors decide on standard deviation of 0.1 and 0.2 for training with Gaussian noise? Can you show an example of clean images, with std=0.1 noise, and with std=0.2 noise in the supplementary information? Please clarify this in the text.
-	The authors state that “Gaussian data augmentation has been widely used to improve the robustness of models in image-related tasks,” yet they only cite one work (ref. 13). Can the authors provide other examples? Otherwise, I suggest revising the wording here.


**Clarity:**

It is correctly stated in the title and throughout the text that the authors are releasing a benchmark, and not a dataset.

**Correctness:**

The authors correctly state in the title and throughout the text that this is a benchmark (not a dataset). The evaluation methods and experiment design are sound, and make sense for the goal which is to assess the robustness of action recognition models.

**Documentation:**

The authors provide a Github link with implementations and code that can be used to create the Mini Kinetics-C and SSV-C benchmarks from their original datasets. It is unclear from the text what the maintenance plan for this codebase is.

**Ethics:**

None that I know of.

**Relation To Prior Work:**

The authors provide a comprehensive list of prior action recognition approaches utilizing both CNNs and Transformers, which are the models used for comparison in their benchmark. Other image-based benchmarks which intend to assess the robustness of image classification models are also discussed (i.e. ImageNet-C).

**Summary And Contributions:**

In this paper, the authors propose a benchmark for evaluating the robustness of spatiotemporal deep learning models against real-world spatial and temporal corruptions. Specifically, they have created subsets of the Kinetics and SSV2 datasets (called Mini Kinetics-C and Mini SSV2-C respectively), and released code for comprehensively simulating various artifacts into the video data. They demonstrate that this benchmark can be used to assess robustness to common video artifacts across various models including CNN and Transformers.

---

> ### Author Response · Authors · 2021-09-29
> **Author Response**
>
> **Q: The authors do not provide model definitions, nor the code used to actually evaluate each model on the Mini Kinetics-C and SSV-C datasets. This limits the reproducibility of the benchmark results in the paper, and the leaderboard found in the Github repository.**
>
> **A:** Thanks for your valuable comments. We have uploaded a framework with structured code examples to our Github repository. Most of the spatial-temporal models can be embedded into this framework for reproducing the benchmark results. Besides, the Github link of each cited spatial-temporal model has been presented in the leaderboards.
>
> In the original Github repository, we have also provided code for researchers to reproduce the corrupted dataset in our benchmark. With the datasets created, they can use their own framework and models for evaluation directly.
>
> **Q: In Section 2.2, I recommend including another work by D Tran, et al. in which video classification networks were further improved by decomposing 3D spatiotemporal convolutions into 2D spatial and 1D temporal convs with an activation between (R(2+1)D networks): D Tran, et al. “A Close Look at Spatiotemporal Convolutions for Action Recognition”**
>
> **A:** Thanks for pointing out this paper. We have added it in Section 2.2 in our revised version.
>
> **Q: How is the frame rate conversion performed exactly? Is the video data re-interpolated? What interpolation scheme is used? Please clarify this in the text.**
>
> **A:** The frame rate conversion is implemented by changing the FPS (frame per second) of videos using FFmpeg tool. The video data will not be interpolated but fewer video frames will be extracted. The target of frame rate conversion is to reduce the number of frames in transmission.
>
> **Q: How did the authors decide on standard deviation of 0.1 and 0.2 for training with Gaussian noise? Can you show an example of clean images, with std=0.1 noise, and with std=0.2 noise in the supplementary information? Please clarify this in the text.**
>
> **A:** The standard deviation of 0.1 is used in the paper [13] to evaluate the corruption robustness of models in the image domain. Because we use similar severity of corruption on each frame as ImageNet-C, we also follow the same setting for Gaussian data augmentation. Furthermore, we have included examples of images with Gaussian noise of std=0.1 and 0.2 in Section 13 of Supplementary.
>
> **Q: The authors state that “Gaussian data augmentation has been widely used to improve the robustness of models in image-related tasks,” yet they only cite one work (ref. 13). Can the authors provide other examples? Otherwise, I suggest revising the wording here.**
>
> **A:** Thanks for your suggestion. We have added 3 more related works in the revised version.

---

> > ### Comment · Reviewer_MVT9 · 2021-10-05
> > **All comments addressed, thanks**
> >
> > Thanks for addressing my comments. My main concern was that the results presented in the benchmark were not reproducible. But the authors latest uploads to their Github repo have directly addressed this concern. I have no further comments and revised my score to a 7 (accept).

---

### Official Review · Reviewer_MJNS · 2021-09-20
**Not clear what the contributions of the benchmark are**

**Rating:** 3
**Confidence:** 4

**Strengths:**

The strengths of this paper is that there is a thorough systematic evaluation of many models (8) on many different corruption types (12). The authors also create two new corrupted datasets - Mini Kinetics-C and Mini SSV2-C - which will be useful for the community to benchmark on moving forward.

**Weaknesses:**

To me, it is not clear what the contributions of this paper are (apart from the synthetic datasets).
The results from the headline experiments - that models with higher clean accuracy have higher corrupted accuracy, and that training for one corruption does not give robustness to other corruptions - have already been studied in the literature in the image classification setting [12, 40]. Thus, is the takeaway simply that "nothing changes in the spatio-temporal modeling setting"? If so, this should be stated more clearly and the paper should be re-organized accordingly.

What I believe would be a more fruitful direction would be understanding what sorts of distribution shifts are likely to arise in video data? The authors acknowledge that [40] find that robustness to synthetic shifts do not imply robustness to natural distribution shifts (which is ultimately what we care about). The presented counterexample from [17] is entirely unconvincing - it is conceivable that being robust to synthetic blurs also gives some robustness to real-world blurs. In general, however, real-world distribution shifts are not so closely aligned with existing synthetic counterparts. For example, videos can vary greatly in subject/object pose, lighting, composition of the scene, weather elements, etc., among many others. Given the comprehensive study of [40] showing that in general synthetic benchmarks do not track real-world use cases, would it not be better to study these natural distribution shifts directly? Or, the authors can disprove previous work such as [40] wrong by collecting data from real-world natural distribution shifts and showing that the proposed synthetic benchmarks track them very closely.

In summary, the major flaw of this work is that it lacks motivation for why synthetic shifts should be considered. I believe this can be remedied by collecting a few examples of natural distribution shifts and showing whether synthetic performance does or does not track it. Additionally, the authors should consider what is different/unique about the spatio-temporal modeling setting (as opposed to image classification) and tailor their analyses, so as not to repeat similar work done previously in image classification.

**Additional Feedback:**

N/A

**Clarity:**

Paper is clear to read. When reading results, I find the rPC hard to interpret (and mostly unnecessary) - looking at the scatter plots such as Figure 2 is far more useful.

**Correctness:**

For the synthetic shifts, why are only half implemented spatially and half implemented temporally? Is there any restriction for coming up with a spatial version of the temporal shifts, and vice versa? If this was possible, one additional interesting analysis would be to see how robust models are to the same shift, but a spatial version vs a temporal version.

Relatedly, it would be interesting to split up figure 2 into spatial corruptions and temporal corruptions and see if the trends line up with each other.

**Documentation:**

There is sufficient documentation.

**Relation To Prior Work:**

Already addressed in weaknesses.

**Summary And Contributions:**

The authors propose a benchmark for video corruption robustness of spatio-temporal models. They implement 12 versions of synthetic corruptions - 6 spatial corruptions and 6 temporal corruptions and subsequently create two corrupted datasets (Mini Kinetics-C and Mini SSV2-C). They find that models with higher clean accuracy also have higher corruption accuracy. They also find training for individual corruptions does not provide robustness to other corruptions.

---

> ### Author Response · Authors · 2021-09-29
> **Author Response to the Weaknesses**
>
> **Q: In summary, the major flaw of this work is that it lacks motivation for why synthetic shifts should be considered. I believe this can be remedied by collecting a few examples of natural distribution shifts and showing whether synthetic performance does or does not track it.**
>
> **A:** Though the paper[40] we cited in our related work found that robustness on common synthetic corruptions does not imply robustness on several types of natural distribution shifts, the natural distribution shifts in this paper only include consistency shift[15][36], dataset shift[1*][2*], and adversarial filtered shift[3*]; None of them consists of the common corruptions introduced in ImageNet-C [18] and our proposed benchmark. Hence, the most recently published paper [17] introduces several complementary types of natural distribution shifts, including real blurry images, which is ignored in [40]. The authors found that synthetic corruptions correlate with corruptions that appear in the wild. Nevertheless, it is evidence that robustness intervention on synthetic robustness benchmark (ImageNet-C [18]) can improve robustness on common corruption arising in nature.
>
> Apart from the image-based robustness benchmarks [18][21][31][43] (cited image tasks papers), synthetic datasets are widely used in different machine learning tasks due to the difficulty of data collection. In the image and video acquisition stage, it is impossible to acquire the ground truth input data in nature in the noise and rain removal area. Most of the research works [1][26][27] apply synthetic noise and rain on clean data for training and evaluation; Similarly, in semantic segmentation, Synthetic Foggy Scene[22] uses synthetic foggy data for training to improve the performance. The Person ReID task, which is related to domain shift, also widely uses synthetic datasets for study [4*][5*][6*][7*].  In the video processing stage, it is common to conduct compression, packet loss, bit error in the lab environments.
>
> Taken together, collecting data from the natural circumstance (the ultimate goal) is ideal, but the difficulty increases exponentially when the scale of datasets and type of corruption increase. As the first move to extend the corruption robustness to video understanding, we aim to understand the impact of common corruption on spatial-temporal models on the basis of the existing large-scale video classification dataset. It enables us to systematically analyze the behavior of the models and shed light on the robustness of spatial-temporal models.
>
> [1*] Objectnet: A large-scale bias-controlled dataset for pushing the limits of object recognition models. NeurIPS 2019
>
> [2*] Do imagenet classifiers generalize to imagenet? ICML 2019
>
> [3*] Natural adversarial examples. CVPR 2021
>
> [4*] Looking Beyond Appearances: Synthetic Training Data for Deep CNNs in Re-identification, Computer Vision and Image Understanding (2018)
>
> [5*] Domain adaptation through synthesis for unsupervised person re-identification, ECCV 2018
>
> [6*] Dissecting Person Re-identification from the Viewpoint of Viewpoint CVPR 2019
>
> [7*] Surpassing Real-World Source Training Data: Random 3D Characters for Generalizable Person Re-Identification, ACMMM 2020
>
> **Q: Additionally, the authors should consider what is different/unique about the spatio-temporal modeling setting (as opposed to image classification) and tailor their analyses, so as not to repeat similar work done previously in image classification.**
>
> **A:** Based on the proposed benchmark, our work explores the generalization and robustness of spatial-temporal models under a general setting. We observed that some trends are similar in both image and video classification, like the impact of capacity, generalization versus robustness, and transferability of robustness. However, beyond the spatial domain, which has been well studied in the image-based tasks, we extend the corruptions to the spatial-temporal level, which considers the correlation of a sequence of frames in video clips. We also explore the model robustness against temporal corruptions specifically. Furthermore, we emphasize the findings especially for spatial-temporal models: (1)Gaussian data augmentation does not improve the robustness of spatial-temporal models; (2)the robustness of the model does not increase with the efficiency of models; (3)the generalization is more related to the robustness of models against temporal corruptions.
>
> Finally, we include a comprehensive study of the input data, training protocol, and model design for spatial-temporal modeling in the original supplementary. We studied the impact of input length, sampling strategy, and 3D module, which are only for spatial-temporal models. This study is more specific for the video classification task.

---

> > ### Comment · Reviewer_MJNS · 2021-09-29
> > **Reviewer response**
> >
> > Thanks for detailed response. After reading the reply thoroughly, I am not convinced by the author response. In my view, the primary reason of a robustness benchmark is to allow for reliable evaluation of ML models in real world settings - that is, if a model does well on a robustness benchmark, then it may reasonably be expected to also be robust to real-world distribution shifts. In fact, this is precisely the motivation the authors reference in the introduction.
> >
> > However, the construction of this benchmark does not inspire any confidence that a model that does well here would also perform well in the real world. As the authors point out, there is a lot of conflicting evidence in the literature to what extent robustness on these synthetic benchmarks transfer to real-world natural distribution shifts. Why is this the case? Well, robustness is *hard* - there are many, many ways the distribution can change in the real world. If we can benchmark models along the appropriate axes - via synthetic proxies - this would provide us a way to test robustness consistently & easily. Of course, the hard question here is figuring out what these appropriate axes are! This difficulty is also why existing literature is conflicted. [40] shows that many "general-purpose" synthetic corruption mechanisms don't transfer over to many real-world distribution shifts. Other examples the authors reference [17][4,6], however, show that specific synthetic modifications can track real-world robustness well. It is worth noting here that these synthetic tasks are successful because they provide targeted modifications to increase robustness (for ex, simulating different clothing [4] or different pedestrian poses [6]). These works have figured out what these "appropriate axes" are that matter for the real world, and then design a synthetic benchmark around that. Without identifying these specific axes, my prior is that any general purpose benchmark is bound to fail as demonstrated in [40].
> >
> > Of course, the easiest way to resolve this question with certainty would be to directly go out and collect data from realistic settings. Of course, this would require significant work, as the authors note - but did we expect to be able to measure robustness without doing the necessary work? If we want to demonstrate that the set of corruptions collected here are reasonable proxies, it seems logical that we should first verify that they are reasonable proxies before suggesting the research community devote countless hours optimizing against this benchmark. In fact, identifying reasonable proxies would provide us much more information than simply another dataset - since real-world data can be hard to understand, knowing a synthetic corruption is a good proxy can potentially inform us a lot about the structure of the collected data.
> >
> > I do feel this paper has significant potential strength and contributions - however, the way it's currently positioned, I cannot rate this paper any higher.

---

> > > ### Author Response · Authors · 2021-09-29
> > > **Author Response**
> > >
> > > Thanks for your reply. Firstly, we would like to emphasize the rationality of proposing 12 types of corruption to evaluate the robustness of spatial-temporal models. Those corruptions come from the video acquisition and processing stages. In the video acquisition, we have provided evidence that synthetic noise[1][31], rain[26][27], fog[22][48], and motion blur[17][20][23] can track the behavior of models in the real world. Besides, the image-based corruption robustness benchmarks[18][21][31][43] use the same corruptions (Motion Blur, Fog, Contrast, Brightness, Saturate, Noise) for evaluation. In video processing, the corruptions are specific to video data, where the compression, bit error, packet loss simulation, frame rate conversion are commonly implemented in the lab environment.
> > >
> > > As we focus on the robust generalization problem instead of specific corruption or distortion, our benchmark is under the out-of-distribution(OOD) generalization scheme [12][18], where the test distribution is unknown and different from the training distribution. As a result, the ‘specific axes’ are hard to be identified in advance but we use different types of corruptions to evaluate the average-case robustness of models. It is a common setting in domain generalization and OOD generalization. Moreover, the synthetic datasets are widely used in domain randomization[2*][5*][6*], domain generalization [3*][4*] and OOD generalization [18][21][31][43][1*].
> > >
> > > When we come back to the discussion of [40] and [17] in such a general setting, it is questionable to claim that [40] evaluated many general-purpose synthetic corruptions or perturbations, because the synthetic distribution shifts only include: Common corruption (general-purpose), Style Transfer and Linf/L2 adversarial attack. The latter two types of shifts are very specific, e.g., there are dozens of adversarial attacks proposed in recent years. [17] evaluated on ImageNet-R and Real Blurry Image, which are two examples showing the robustness intervention on ImageNet-C can improve the robustness of models on both synthetic and natural distribution shifts. These two works are not conflicting but complementary with each other. We agree that robustness is hard and multivariate, it requires us to study from not only one aspect.
> > >
> > > Hence, we make a move in the spatial-temporal domain to understand the sustainability of models under common corruptions approximately. The benchmark study is not able to explain all the circumstances but at least sheds light on the corruption robustness of spatial-temporal models.
> > >
> > > [1*]Invariant risk minimization. arXiv preprint arXiv:1907.02893, 2019.
> > >
> > > [2*] Training Deep Networks with Synthetic Data: Bridging the Reality Gap by Domain Randomization, CVPRW2018
> > >
> > > [3*]The synthia dataset: A large collection of synthetic images for semantic segmentation of urban scenes, CVPR2016
> > >
> > > [4*]Playing for data: Ground truth from computer games, ECCV2016
> > >
> > > [5*] Domain Randomization and Pyramid Consistency: Simulation-to-Real Generalization without Accessing Target Domain Data, ICCV2019
> > >
> > > [6*]Structured Domain Randomization: Bridging the Reality Gap by Context-Aware Synthetic Data, ICRA2020

---

> ### Author Response · Authors · 2021-09-29
> **Author Response to the Correctness**
>
> **Q: For the synthetic shifts, why are only half implemented spatially and half implemented temporally? Is there any restriction for coming up with a spatial version of the temporal shifts, and vice versa? If this was possible, one additional interesting analysis would be to see how robust models are to the same shift, but a spatial version vs a temporal version.**
>
> **A:**  We would like to clarify that the corruptions are not half-implemented spatially and half-implemented temporally. As defined in our introduction section, spatial corruptions only depend on the content of a single frame while independent from the other frames. We made an assumption that spatial corruptions like noise, rain are i.i.d in each frame. In contrast,  temporal corruptions highly correlate to a continuous sequence of frames. In the spatial dimension, it may cause spatial distortion in the frame; in the temporal dimension, the spatial distortion may propagate to a sequence of frames. For example,  the distortion caused by packet loss in the previous frame will augment in the following frames.
>
> Hence, there is a restriction to generate a spatial version of temporal corruption, which requires manipulating the corruptions on purpose.

---

### Official Review · Reviewer_NXNd · 2021-09-22

**Rating:** 8
**Confidence:** 3
**Clarity:** The paper is very well written.

**Strengths:**

* This is a good benchmark. The paper performs corruption on two of the very popular action recognition datasets – Kinetics and Something Something V2
* Multiple types of corruptions can be performed at different levels.
* Well written paper with extensive analysis and discussion for temporal and spatial corruptions.
* The contribution seems to be significant in the field of action recognition and video understanding.


**Weaknesses:**

No major weaknesses that are not already accounted for in the paper. In fact, the authors include a refreshingly up-front discussion of their results. Potentially the authors can throw some light on the computation required since action recognition datasets tend to be massive and computationally hungry.

**Additional Feedback:**

It would be interesting to analyze which actions particularly fail on each dataset and like wise for each model.

**Correctness:**

They seem correct to me.



**Documentation:**

Yes the GitHub seems to be well documented.

**Ethics:**

The dataset is taken from existing datasets so none.

**Relation To Prior Work:**

Its relation to prior works is clearly discussed.



**Summary And Contributions:**

The paper proposes a benchmark on activity recognition dataset to measure robustness against the spatial and temporal corruptions. The paper establishes Mini Kinetics-C and Mini SSV2-C for the evaluation purposes. This is the first paper that evaluates corruption robustness of spatial-temporal video models.

---

> ### Author Response · Authors · 2021-09-29
> **Author Response**
>
> **Q: Potentially the authors can throw some light on the computation required since action recognition datasets tend to be massive and computationally hungry.**
>
> **A:** Thank you for your invaluable comments. We agree that action recognition datasets are massive, and the computational cost is very high for evaluating spatial-temporal models. To enable researchers to explore the impact of corruption on models, we provided a smaller version of Mini Kinetics-C with 40 classes for training and evaluation, as suggested in Section 5.2.

---

### Decision · Program_Chairs · 2021-10-09

**Decision:**

Accept

**Comment:**

This paper presents a novel benchmark for evaluating robustness against spatio-temporal video corruptions. While the reviewers generally agree that the paper addresses a novel important question, it received split reviews of (3,8,6,7) where the main concern raised by one reviewer is the lack of real corruption data to prove that the proposed synthetic dataset really aligns with the real-world corruption cases.
The reviewers and AC had an intensive and fruitful discussion on this point. Importantly, we all agree that the concern totally makes sense and we all have to be aware of this fundamental issue. At the same time, we also understand that sometimes collecting ideal data is quite difficult and demanding, and we need to find out a reasonable compromise to go forward step by step. From this viewpoint, three reviewers clearly support that the paper opens up a novel research direction and has enough contributions to be published in its current form. The AC considers their conclusion is valid and would like to recommend acceptance.
In addition to that, as a part of NeurIPS, papers in this track are expected to lead this kind of fundamental and scientific discussion in terms of data, so I'd like to encourage authors to consider this issue thoroughly and enhance the discussion properly (in particular, the limitation of the dataset).